



**Numerical analysis of the impact of agricultural emissions on PM$_{2.5}$ in China using a**
**high-resolution ammonia emissions inventory**
Xiao Han[1,2], Lingyun Zhu[5], Mingxu Liu[4], Yu Song[4] Meigen Zhang[1,2,3],
[1]*State Key Laboratory of Atmospheric Boundary Layer Physics and Atmospheric Chemistry, Institute of*
*Atmospheric Physics, Chinese Academy of Sciences, Beijing 100029, China*
[2]*College of Earth and Planetary Sciences, University of Chinese Academy of Sciences, Beijing 100049,*
*China*
[3]*Center for Excellence in Urban Atmospheric Environment, Institute of Urban Environment, Chinese*
*Academy of Sciences, Xiamen 361021, China*
[4]*State Key Joint Laboratory of Environmental Simulation and Pollution Control, Department of*
*Environmental Science, Peking University, Beijing 100871, China.*
[5]*Shanxi Province Institute of Meteorological Sciences, Taiyuan 030002, China*
Corresponding author:
Lingyun Zhu
Shanxi Province Institute of Meteorological Sciences
Xinjian Road 65#, Taiyuan, Shanxi province, China
Post code: 030002
Tel: 86-0351-4077738
Fax: 86-0351-4077738
E-mail: zhlyun@126.com
Meigen Zhang
LAPC, Institute of Atmospheric Physics, Chinese Academy of Sciences
HuaYanBeiLi 40#, Chaoyang District
Beijing, China
Post code: 100029
Tel: 86-010-62379620
Fax: 86-010-62041393
E-mail: mgzhang@mail.iap.ac.cn



**Abstract**
China is one of the largest agricultural countries in the world. The $NH_3$ emissions from agricultural activities
in China significantly affect regional air quality and horizontal visibility. To reliably estimate the influence
of $NH_3$ on agriculture, a high-resolution agricultural $NH_3$ emissions inventory, compiled with a 1 km × 1
km horizontal resolution, was applied to calculate the $NH_3$ mass burden in China. The key emission factors
of this inventory were enhanced by considering the results of many native experiments, and the activity
data of spatial and temporal information were updated using statistical data from 2015. Fertilizer and
husbandry, as well as farmland ecosystems, livestock waste, crop residue burning, fuel wood combustion,
and other $NH_3$ emission sources were included in the inventory. Furthermore, a source apportionment tool,
ISAM (Integrated Source Apportionment Method), coupled with the air quality modeling system RAMS-
CMAQ (Regional Atmospheric Modeling System and Community Multiscale Air Quality), was applied to
capture the contribution of $NH_3$ emitted from total agriculture (Tagr) in China. The aerosol mass
concentration in 2015 was simulated, and the results showed that a high mass concentration of $NH_3$, which
exceeded 10 μg m$^{-3}$, appeared mainly in the North China Plain (NCP), Central China (CNC), the Yangtz
River Delta (YRD), and the Sichan Basin (SCB), and the annual average contribution of Tagr $NH_3$ to $PM_{2.5}$
mass burden in China was 14-18%. Specific to the $PM_{2.5}$ components, Tagr $NH_3$ provided a major
contribution to ammonium formation (87.6%) but a tiny contribution to sulfate (2.2%). In addition, several
brute-force sensitivity tests were conducted to estimate the impact of Tagr $NH_3$ emissions reduction on the
$PM_{2.5}$ mass burden. Compared with the results of ISAM, it was found that even though the Tagr $NH_3$ only
contributed 10.1% of nitrate under current emissions scenarios, the reduction of nitrate could reach 98.8%
upon removal of the Tagr $NH_3$ emissions. The main reason for this deviation could be that the $NH_3$
contribution to nitrate is small under "rich $NH_3$" conditions and large in "poor $NH_3$" environments. Thus,
the influence of $NH_3$ on nitrate formation could be enhanced with the decrease of ambient $NH_3$ mass
concentration.










## 1. Introduction

Ammonia ($NH_3$) is an important pollution species which principal neutralizing agent for the acid aerosols, $SO_4^{2-}$ and $NO_3^-$ formed from the $SO_2$ and $NO_x$ (Chang, 1989; McMurry et al.; 1983). In addition, $NH_3$ also influences the rate of particle nucleation (Ball et al.; 1999; Kulmala et al.; 2002) and enhances secondary organic aerosols (SOA) yields (Babar et al.; 2017). The widespread haze events have frequently occurred in most regions of eastern China in recent years, and several studies have reported that the secondary inorganic salts, including sulfate, nitrate, and ammonium, were the majorities of the total aerosols in the urban and rural regions (Tao et al.; 2014; Wang et al.; 2016; Zhang et al.; 2012; Lai et al.; 2016; Zhang et al.; 2018). Therefore, besides the heavy emissions of $SO_2$ and $NO_2$, $NH_3$ emissions from the agriculture activities are also non-negligible.

China is one of the largest agricultural countries in the world. Even though the annual emissions budget of $NH_3$ decreased from 2006 to 2012, the emissions were still high and reached 9.7-12 Tg (Kang et al., 2016; Xu et al., 2016; Zhou et al., 2015), leading to high ambient $NH_3$ concentrations. These massive $NH_3$ levels significantly affect regional air quality and horizontal visibility. Firstly, the major $PM_{2.5}$ components, $(NH_4)_2SO_4$, $(NH_4)_3H(SO_4)_2$, $NH_4HSO_4$, and $NH_4NO_3$ were partially or fully produced from the neutralization of $H_2SO_4$ and $HNO_3$ by the reaction with $NH_3$ (Tanner et al.; 1981; Brost et al.; 1988; Quan et al.; 2014; Zhao et al.; 2013; Zhang et al.; 2014). Studies also showed that $NH_3$ improves the $H_2SO_4$ nucleation by 1-10 times (Benson et al.; 2011), and provides sufficient new particle to alter the number and size distributions. Thus, the $NH_3$ and its secondary product $NH_4^+$ play an important role in the formation of air pollution and haze days. Research has shown that approximately 80% of total anthropogenic $NH_3$ emissions derived from agricultural sources and livestock manure provided a greater contribution than synthetic fertilizer (Kang et al., 2016; Zhou et al., 2016). The Chinese government has undertaken several control strategies to reduce particulate pollution and its precursors, such as catalytic reduction systems in the power sector (Xia et al., 2016), measures to change coal to gas for residential life and heating (Ren et al., 2014), etc. Related observations have shown that the mass burdens of $SO_2$ and $NO_x$ have decreased distinctly in recent years (De Foy et al., 2016; Wang et al., 2015; Zheng et al., 2018). However, no specific measures for agricultural $NH_3$ emissions control have been implemented to date, and the total agricultural $NH_3$ emissions budget did not change substantially from 2010 to 2017 (Zheng et al., 2018).

In addition, accurate information on agricultural $NH_3$ emissions is also important for estimating the $NH_3$ mass burden and its environmental effect. There have been several studies focusing on $NH_3$ emissions



from agricultural activities in China or East Asia. REAS (Regional Emission inventory in Asia) version 2
established an anthropogenic emissions inventory that included the source of agricultural $NH_3$ (fertilizer
application and livestock) (Kurokawa et al.; 2013). This inventory, targeting years from 2000 to 2008, has
a $0.25° \times 0.25°$ spatial resolution with monthly variation. MASAGE_$NH_3$ (Magnitude and Seasonality of
Agricultural Emissions model for $NH_3$) was used to develop a bottom-up $NH_3$ emissions inventory by using
the adjoint of the GEOS-Chem chemical transport model (Paulot et al.; 2014). The inverse of the network
data for $NH_4^+$ wet deposition fluxes from 2005-2008 was used to optimize the $NH_3$ emissions in China in
this inventory. Fu et al. (2015) used the CMAQ (Community Multiscale Air Quality) model coupled to an
agro-ecosystem, which could obtain hourly emissions features by online model calculation, to estimate $NH_3$
emissions in 2011 with high spatial and temporal resolution. These $NH_3$ emissions inventories provided
very useful datasets for understanding the distribution features of the $NH_3$ mass burden in China. However,
with population migration, economic growth, and the increased consumption of agricultural products, the
spatial distribution and strength of agricultural $NH_3$ emissions has significantly changed in China during
the last decade (Xu et al., 2017), so that reliable emissions information based on recent years is also
necessary for estimating the $NH_3$ mass burden.
Previous studies have investigated the influence of $NH_3$ emissions on aerosol loading in several typical
areas of China. Wu et al. (2008) conducted sensitivity studies to assess the impact of livestock-produced
$NH_3$ emissions on $PM_{2.5}$ mass concentration in North China by using the MM5/CMAQ modeling system.
The results showed that the livestock-produced $NH_3$ provided >20% contributions to nitrate and ammonium,
but provided only a small contribution to sulfate. Wang et al. (2011) used the response surface modeling
technique to estimate the contribution of $NH_3$ emissions in East China and found that total $NH_3$ emissions
contributed 8-11% to $PM_{2.5}$ concentration, and the nonlinear effects were significant while the transition
between $NH_3$ rich and poor conditions. Fu et al. (2017) and Zhao et al. (2017) also investigated the impact
of $NH_3$ emissions on $PM_{2.5}$ in East China and the Hai River Basin. However, the related studies were few
and focused mainly on local regions; furthermore, most of them generally used the brute-force sensitivity
method to estimate the $NH_3$ impact based on the chemistry model, which reflected the change in particulate
concentration with emissions reduction (Koo et al., 2009).
PKU-$NH_3$, a comprehensive high-resolution $NH_3$ emissions inventory based on the year 2015, was
applied in this study to capture the agricultural $NH_3$ mass concentration in China, and the contribution to
$PM_{2.5}$ particles was estimated with an RAMS-CMAQ air quality modeling system, coupled with the online



source tagged module ISAM. Compared with previous studies, this high-resolution agricultural NH₃
emissions inventory was more accurate and reflected the latest spatial and temporal distribution features
(Liu et al.; 2019). The major trace gases and aerosol species in 2015 were simulated by the modeling system
and evaluated by several observational data. The contribution to the pollutant concentrations was tagged
and quantified by RAMS-CMAQ-ISAM under the current scenario (Wang et al., 2009). Then, several brute-
force sensitivity tests were conducted to estimate the effect of reducing agricultural NH₃ emissions on the
$PM_{2.5}$ mass burden. The results from the source apportionment simulation and brute-force sensitivity tests
in January, April, July, and October are presented here, and the detailed features over seven major populated
areas of China (as shown in Figure 1) are discussed.

**2. Methodology**
The emissions inventory can be described as follows: First, the NH₃ emissions data in China were
provided by the PKU-NH₃ emissions inventory (Kang et al., 2016; Zhang et al., 2018). This inventory was
developed on the basis of previous studies (Huang et al., 2012) and improved the horizontal resolution and
accuracy. It was compiled at a 1 km × 1 km horizontal resolution, with monthly statistical data from 2015.
Some of the most uncertain parameters, the emission factors applied in this inventory, were enhanced by
considering as many native experiment results as possible, with ambient temperature, soil acidity, and
changes in other factors. In addition, this inventory not only included fertilizer and husbandry emissions
from agricultural activities but also collected the emissions data of farmland ecosystems, livestock waste,
biomass burning (forest and grassland fires, crop residue burning, and fuel wood combustion), and other
sources (excrement waste from rural populations, the chemical industry, waste disposal, NH₃ escape from
thermal power plants, and traffic sources). Second, the anthropogenic emissions of primary aerosols and
their precursors were obtained from the MIX Asian emission inventory (base year 2012), prepared by the
Model Inter-Comparison Study for Asia (MICS-ASIA III) (Lu et al., 2011; Lei et al., 2011). The
anthropogenic emissions sources of $SO_2$, $NO_x$, volatile organic compounds (VOCs), black carbon (BC),
organic carbon (OC), primary $PM_{2.5}$, and $PM_{10}$ were obtained from the monthly MIX inventory, with a 0.25°
× 0.25° spatial resolution. The REAS (Regional Emission Inventory in Asia; Version 2; Kurokawa et al.,
2013) and GFED (Global Fire Emissions Database; Version 3; van der Werf et al., 2010) were used to
provide the VOCs, and nitrogen oxides from flight exhaust, lightning, paint, wildfires, savanna burning,
and slash-and-burn agriculture.



The RAMS-CMAQ modeling system was applied to simulate the transformation and transport of
pollutants in the atmosphere. The CMAQ regional air quality model (version 5.0.2) released by the US
Environmental Protection Agency (Eder et al., 2009; Mathur et al., 2008) was a major component of the
RAMS-CMAQ modeling system. In this model, the CB05 (version CB05tucl) chemical mechanism
(Whitten, 2010) and the sixth-generation CMAQ aerosol model (AERO6) were used to treat the gas-phase
chemical mechanism and the formation and dynamic processes of aerosols. The ISORROPIA model
(version 2.1) (Fountoukis and Nenes, 2007) was used to describe the thermodynamic equilibrium of gas-
particle transformation. The highly versatile RAMS numerical model, which can well capture the boundary
layer and the underlying surface, was applied to provide the meteorological fields for CMAQ (Cotton et al.,
2003). The European Centre for Medium-Range Weather Forecasts reanalysis datasets ($1° \times 1°$ spatial
resolution) were used to supply the background fields and sea surface temperatures. The model domain
(Figure 1) was 6654 km $\times$ 5440 km, with 64 km$^2$ fixed-grid cells, and a rotated polar stereographic map
projection covering the entire mainland of China and its surrounding regions was used. The model had 15
vertical layers, and half of them were located in the lowest 2 km to provide a more precise simulation of
the atmospheric boundary layer. Several previous studies have demonstrated that the modeling system
performs well when simulating the spatial and temporal distribution of China's major aerosol components
(Han et al., 2013, 2014, 2016).
The ISAM is a flexible and efficient online source apportionment implementation, which was used to
track multiple pollutants emitted from different geographic regions and source types. Compared with its
previous version TSSA (Tagged Species Source Apportionment), the processes of tracking tagged tracer
transport and precursor reactions were optimized for balancing the computational requirements and reliable
representation of the physical and chemical evolution. To reduce the nonlinear effect during phase
transformation and relative chemical interactions, a standalone subroutine "wrapper" approach was applied
to the ISAM model to apportion the secondary PM species and their precursor gases during the
thermodynamic equilibrium simulation; a hybrid approach, which employed the LU decomposition
triangular matrices (Yang et al., 1997), was also developed for describing the gas-phase chemical
interactions. In this study, ISAM was coupled with RAMS-CMAQ and was set to trace the transport and
chemical reactions of NH$_3$ from fertilizer and husbandry emissions sectors to quantitatively estimate the
contribution of agricultural NH$_3$ emissions to the PM$_{2.5}$ mass concentration in China.



## 3. Model evaluation


To evaluate the model performances, several observation data were compared with the simulation

results. The meteorological factors are important to capture the formation processes and transport of
secondary aerosols. Thus, in this paper, the observed meteorological data from surface stations of the
Chinese National Meteorological Center are collected to evaluate the performance of the model. The detail
information is described in Appendix A. Furthermore, the observed $SO_2$, $NO_2$, and $PM_{2.5}$ released from the
Ministry of Environmental Protection of China were applied to evaluate the modeled mass concentration
of these pollutants. The observation data at 416 stations, located in 101 model grids (distributed in Beijing,
Tianjin, Hebei, Shandong, Shanxi, Henan, Jiangsu, and Anhui), were collected, and the values in same grid
were averaged. The scatter plots of comparison are shown in Figure 2, and the statistical parameters
between the observations and simulations are listed in Table 1. It can be seen that most of the scatter points
broadly gather around the 1:1 solid line. The correlation coefficients in this table are all higher than 0.5,
which indicates that the model can capture the regional variation in the measurements. The standard
deviations between the observations and simulations were similar in most cases, except for $SO_2$ in January.
The largest deviation of the modeled mean, which was higher than that of the observation, was also between
the observed and modeled $SO_2$ in January. However, the correlation coefficients reached 0.71 in January,
and the performance of the model in other months was relatively good, as shown in Table 1. It can be
deduced that the obvious deviation may be a systemic underestimation due to the lack of emission intensity
in this month.

The horizontal distributions of modeled monthly $NH_3$ mass concentration in January, April, July, and

October in 2015 are shown in Figure 3. Pan et al. (2018) provided the distributions of satellite $NH_3$ total
column distribution and the surface $NH_3$ concentrations at several observation sites (as shown in Figure 1
in the aforementioned study). As shown from their results, the highest mass burden was concentrated
mainly in the North China Plain (NCP), Central China (CNC), the Yangtz River Delta (YRD), and the
Sichan Basin (SCB). The simulation results in this study broadly reflected these distribution features. The
$NH_3$ concentrations in these regions reached 10-25 μg m$^3$ in Pan et al. (2018), which also coincided well
with the simulation results. However, some obvious deviations appeared in the eastern part of Gansu
province. The modeled $NH_3$ in these regions was slightly higher than that of the observations in Pan et al.
(2018). Zhang et al. (2018) also showed the $NH_3$ mass concentration in four seasons over China from
simulation (horizontal distribution) and ground-based measurements (point values) in Figure 9 of their



study. Aside from the regions mentioned in Pan et al. (2018), the high mass burden of $NH_3$ also appeared

in the NEC, as shown by both simulation and observation results in Zhang et al. (2018). Generally, this

distribution feature should be reasonable because the Three River Plain located in NEC is an important

agriculture base in China, and the $NH_3$ emissions in this region can be strong during spring and summer.

The simulation results in this study also supported the seasonal variation of the $NH_3$ mass burden shown in

Zhang et al. (2018), which was higher in summer and lower in winter, and the magnitudes of the two were

close. Thus, it can be seen that the $NH_3$ concentration modeled by RAMS-CMAQ was reliable and can be

applied to the analysis in this study.

## 4. Results and discussions

The horizontal distributions of modeled monthly $PM_{2.5}$ mass concentrations in January, April, July,

and October 2015 are shown in Figure 4. Over the eastern part of China, the heavy $PM_{2.5}$ pollution occurred

in January, and the relatively better air quality appeared in July. The large $PM_{2.5}$ mass burden, exceeding

200 μg m$^3$ in January, was mainly concentrated in the NCP, the Yangtze River Valley of CNC, and the SCB,

which broadly coincided with the regions covered by a high mass burden of $NH_3$, as shown in Figure 3. In

addition, the $PM_{2.5}$ mass burden (50-150 μg m$^{-3}$) was obviously lower in July than in the other months.

Since $NH_3$ concerns mainly with secondary inorganic aerosols: sulfate, nitrate, and ammonium (SNA)

formation, the analysis hereafter will mainly focus on the SNA. Figure 5 presents the modeled monthly

SNA mass concentrations in January, April, July, and October 2015. The mass loading of SNA generally

contributed 40-60% of the total $PM_{2.5}$ in the eastern part of China, which was comparable with previous

studies (Cao et al., 2017; Chen et al., 2016; Lai et al., 2016; Wang et al., 2016). The distribution pattern and

seasonal variation of SNA also followed the features of $PM_{2.5}$, and the high mass concentration of SNA

exceeded 100 μg m$^{-3}$ in January.

Then, the contributions of $NH_3$ from multiple agricultural emissions (including fertilizer, husbandry,

farmland ecosystems, livestock waste, crop residue burning, and excrement waste from rural populations)

to aerosols were calculated using RAMS-CMAQ-ISAM; the monthly average contribution percentage of

total agriculture activities (Tagr) in January, April, July, and October are shown in Figure 6. Generally, Tagr

$NH_3$ provided a 30-50% contribution to the SNA over most of eastern China in January and October, and a

20-40% contribution in April and July. The relatively lower value appeared mainly in April. The regional

and annual average percent contributions of Tagr to sulfate, nitrate, ammonium, SNA, and $PM_{2.5}$ are shown



in Table 2. As shown in this table, Tagr $NH_3$ provided the major contribution to ammonium, which reached
approximately 90%, and a relatively small contribution to nitrate mass burden, which was 5-10%. However,
the contribution to sulfate was tiny, and the main reason is that there are various methods of sulfate
formation from $SO_2$ other than neutralization by $NH_3$, such as oxidation by $H_2O_2$, $O_3$, or peroxyaceticn acid.
Tagr $NH_3$ provided a 28-37% contribution to the SNA mass concentration, and the spatial features of the
Tagr $NH_3$ contribution to the $PM_{2.5}$ mass concentration were similar to the features of SNA. Generally, it
provided an approximately 14-18% contribution to the total $PM_{2.5}$ mass concentration in these places, and
the largest annual average contribution appeared in CNC (17.5%).
In addition, the brute-force method (zero-out sensitivity test), which can capture the effect of emissions
changes on aerosol mass burden, was applied to investigate the impact of the removal of Tagr $NH_3$
emissions. Unlike online source apportionment, the brute-force method mainly reflects the disparity of the
chemical balance caused by the emissions change, which could significantly alter secondary pollutant
formation. Several sensitivity tests were conducted, and the results are shown in Figure 7 and Table 3.
Figure 7 presents the mass burden variation of SNA associated with Tagr $NH_3$ removal. From Figure 7, it
can be seen that the reduction patterns of the aerosol broadly followed those of their mass burden. The
significant reduction of SNA mainly appeared in the high concentration regions, and generally exceeded
25 $\mu g\ m^{-3}$. Table 3 shows the percentage of the variation of sulfate, nitrate, ammonium, SNA, and $PM_{2.5}$.
Compared with Table 2, it can be seen that the variation percent of SNA and $PM_{2.5}$, which reached 40-51%
and 23-35%, respectively, were approximately two times higher than those of the contribution percent, and
this significant distinction was mainly caused by the variation of nitrate: the contribution of Tagr $NH_3$ to
nitrate was generally below 10%, as shown in Table 2, but the reduction of nitrate associated with removing
Tagr $NH_3$ emissions could exceed 95%, as shown in Table 3. This difference between the results of ISAM
and brute-force was expected as a result of high nonlinearity in the $NO_x$ chemistry. The nitrate formation
could become more sensitive when the "rich $NH_3$" environment shifts to a "poor $NH_3$" environment, which
means the decrease of the nitrate mass burden would accelerate with the $NH_3$ emissions reduction.
Therefore, it can be deduced that the contribution of $NH_3$ to nitrate should be significantly lower under a
"rich $NH_3$" environment than that under a "poor $NH_3$" environment. A similar phenomenon was also
reported in previous studies (Wang et al., 2011; Xu et al., 2016). To prove this point, further brute-force
sensitivity tests were conducted. The variations of sulfate, nitrate, ammonium, and SNA mass burden
associated with the reduction of $NH_3$ emissions (80%, 50%, 40%, 30%, 20%, and 10% TA $NH_3$ emission,



respectively) is shown in Figure 8. It can be seen that the decrease in nitrate mass concentration was more
rapid than that of ammonium, and the trend became slightly faster with the reduction of NH$_3$ emissions
(signifying the transition from a "rich NH$_3$" to a "poor NH$_3$" environment) in the regions with a high mass
burden of NH$_3$: BTH, NEC, SCB, and SDP. Furthermore, this acceleration stopped while 20% of NH$_3$
emissions remained.

**5. Conclusions**
The emission budget of agriculture NH$_3$ was huge and played an important role on the regional particle
pollution in China. As a precursor of the secondary aerosol, reasonably estimate the nonlinear processes of
secondary aerosol formation should be the key point for capturing the contribution of NH$_3$ to particle
pollution. In this study, the air quality modeling system RAMS-CMAQ was applied to simulate spatial-
temporal distribution of trace gas and aerosols in 2015. In addition, the PKU-NH$_3$ emission inventory which
compiled on 1km×1km horizontal resolution with monthly based data was applied to accurately capture
the agriculture NH$_3$ emission features in China. Then, the source apportionment module ISAM was coupled
into this modeling system to quantitatively estimate the contribution of agriculture NH$_3$ to PM$_{2.5}$ mass
burden. The brute-force sensitivity tests were also conducted for discussing the impact of the agriculture
NH$_3$ emission reduction. The meteorological factors and mass concentration of NH$_3$, SO$_2$, NO$_2$, and PM$_{2.5}$
from simulation were evaluated and showed well agreement with the observation data. Some interesting
results were explored and summarized as follow:
(1) The high mass burden of NH$_3$ could exceeded 10 μg m$^{-3}$, and mainly appeared in the NCP, CNC,
YRD, and SCB. These regions were highly coincidence with the regions that heavy particle pollution
covered in China. Therefore, it can be deduced that the influence of agriculture NH$_3$ on the PM$_{2.5}$ mass
concentration should be significant.
(2) The results from ISAM simulation shows that the Tagr NH$_3$ provided 14-18% contribution to the
PM$_{2.5}$ in the most part of east China, and the largest annual average contribution appeared in CNC (17.5%).
Specific to the SNA components, the annually and regional average contribution of Tagr NH$_3$ to ammonium,
nitrate, sulfate was 87.6%, 10.1%, and 2.2% in China. The agriculture NH$_3$ emission provided major
contribution to the ammonium formation, but tiny contribution to the sulfate due to the various other ways
of sulfate formation.
(3) The brute-force sensitive test could reflect the effect of changing Tagr NH$_3$ emission on PM$_{2.5}$ mass



burden. The results indicated that the reduction percent of PM$_{2.5}$ mass burden due to removal Tagr NH$_3$
emission could reach 23-35% in the most part of east China, which was approximately two times higher
than the contribution. The reduction percent of nitrate that reached exceed 95% was the main reason caused
this significant different. In addition, the further analysis proved that the ambient NH$_3$ mass burden could
obviously affects its contribution to the SNA formation: the NH$_3$ contribution to nitrate should be lower
under "rich NH$_3$" and higher under "poor NH$_3$". Therefore, the influence of NH$_3$ would enhance with the
decreasing of ambient NH$_3$ mass concentration.
It is suggested that the influence of NH$_3$ on the PM$_{2.5}$ mass burden is complex because of the
nonlinearity of secondary aerosol formation. Significant deviation exists between the results from ISAM
and the brute-force method; therefore, these two kinds of results should be distinguished and applied to
explain different issues: the contribution under the current scenario and the effect due to emissions reduction,
respectively. The modeling system is a versatile tool that allows us to investigate this valuable information
to choose more efficient strategies for reducing the impact of agricultural NH$_3$ and improving air quality.

**Acknowledgments**
This work was supported by the Strategic Priority Research Program of the Chinese Academy of
Sciences (XDA19040204), and the National Natural Science Foundation of China (41830109).















**Appendix A**


The daily average temperature, relative humidity, wind speed and maximum wind direction in January
and July 2015 were compared with the surface shared data from the Chinese National Meteorological
Center (http://data.cma.cn/) in 9 stations. The comparison results are shown in Figure A1-A4. These stations
are located in the East China where the high $NH_3$ emission regions. Generally, the modeled temperature
was in good agreement with the observed data, and can reflect the large fluctuation and seasonal variation
of relative humidity as well, except that some of the extreme high or low values appeared abruptly. As
shown in Figure A3, most of the daily average wind speed was lower than 3 m s$^{-1}$ at Zhengzhou, Jinan,
Miyun, and Baoding station (all located in the North China Plain), which means the diffusion condition was
not good due to the stable weather. Otherwise, the relatively strong wind appeared at Nanjing, Chaoyang,
Nanning, and Tianjin. The modeled wind speed generally reproduced all these features. The direct
comparison between observed and modeled wind direction which can be easily influenced by the
surrounding surface features is difficult. Nevertheless, the prevailing wind direction in different seasons
can be captured by the simulation results for all stations.


















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



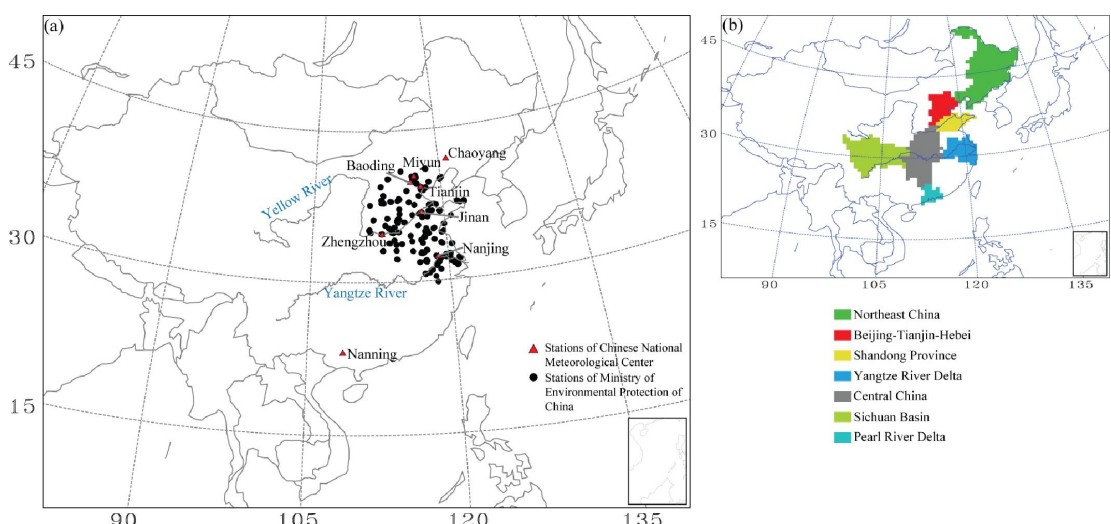


Figure 1. Model domain used in this study and the geographic locations of Northeast China, Beijing-Tianjin-Hebei,
Shandong Province, Yangtze River Delta, Central China, Sichuan Basin, and Pearl River Delta. The location of
observation data was also shown in the model domain.



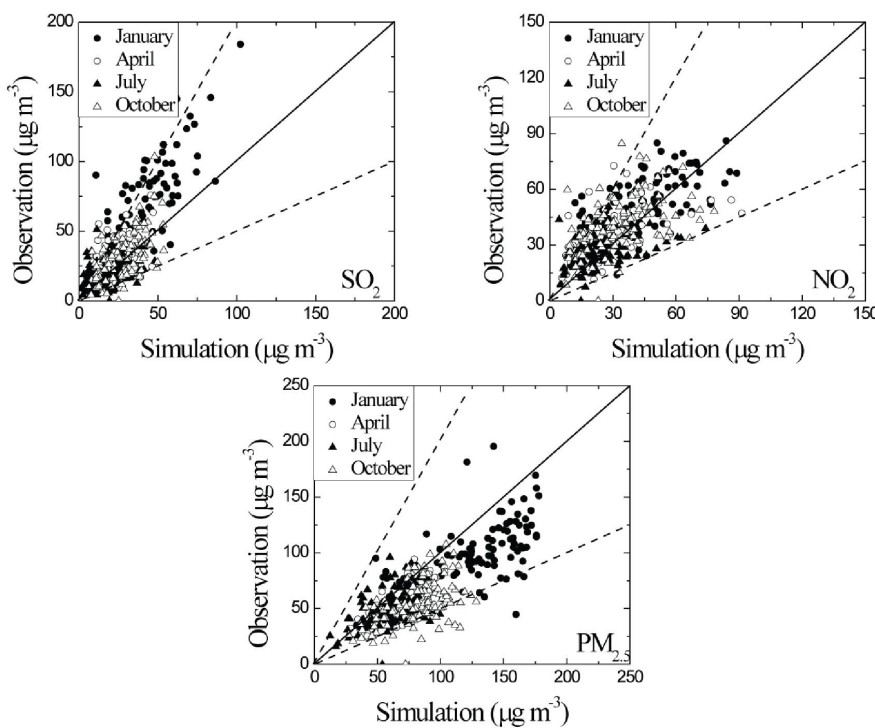


Figure 2. The scatter plots between the modeled and the observed monthly $SO_2$, $NO_2$, and $PM_{2.5}$ in 2015. The solid lines
are 1:1 and the dashed lines are 2:1 or 1:2.


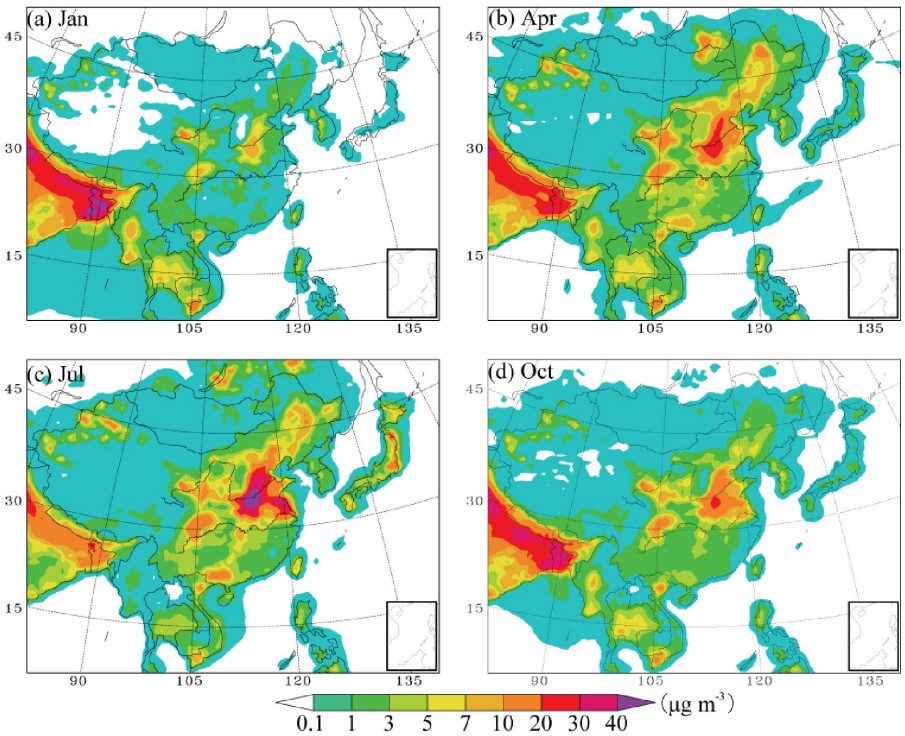

Figure 3. The horizontal distributions of the modeled monthly NH$_3$ mass concentration in January, April, July, and
October in 2015.

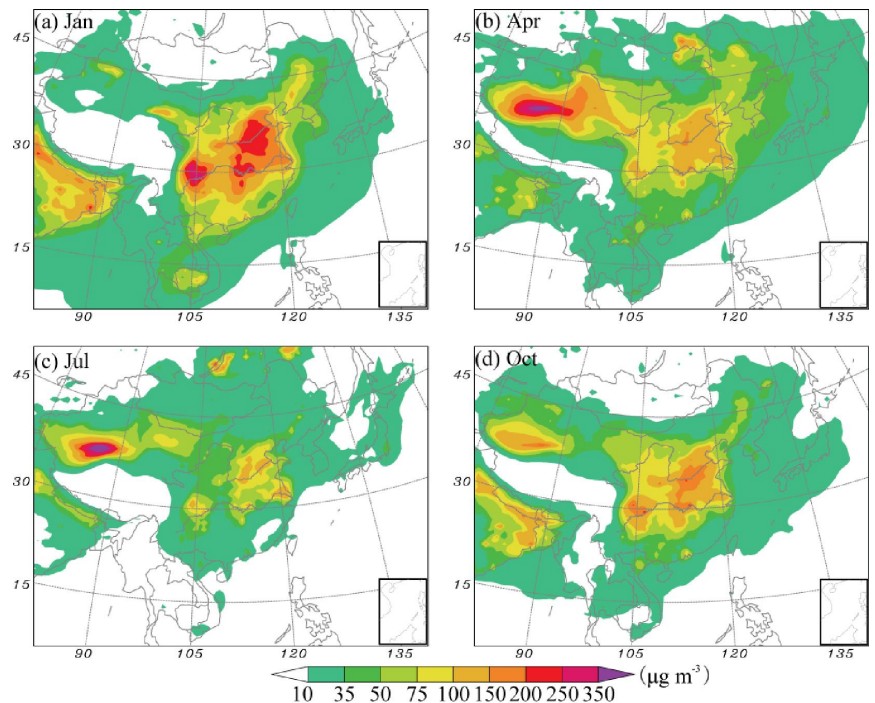

Figure 4. The horizontal distributions of the modeled monthly PM$_{2.5}$ mass concentration in January, April, July, and October in 2015.

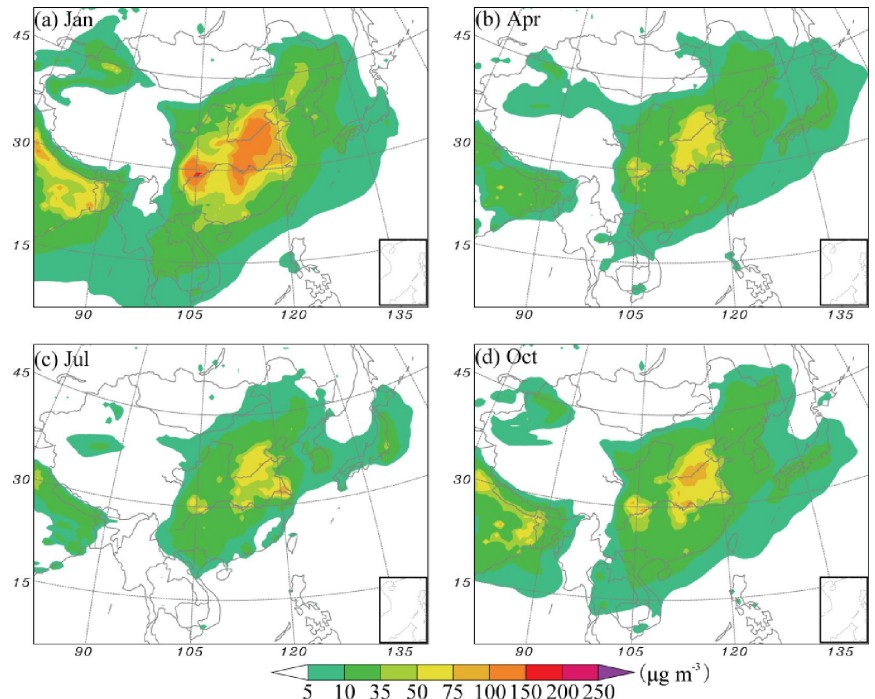


Figure 5. The horizontal distributions of the modeled monthly SNA mass concentration in January, April, July, and
October in 2015.


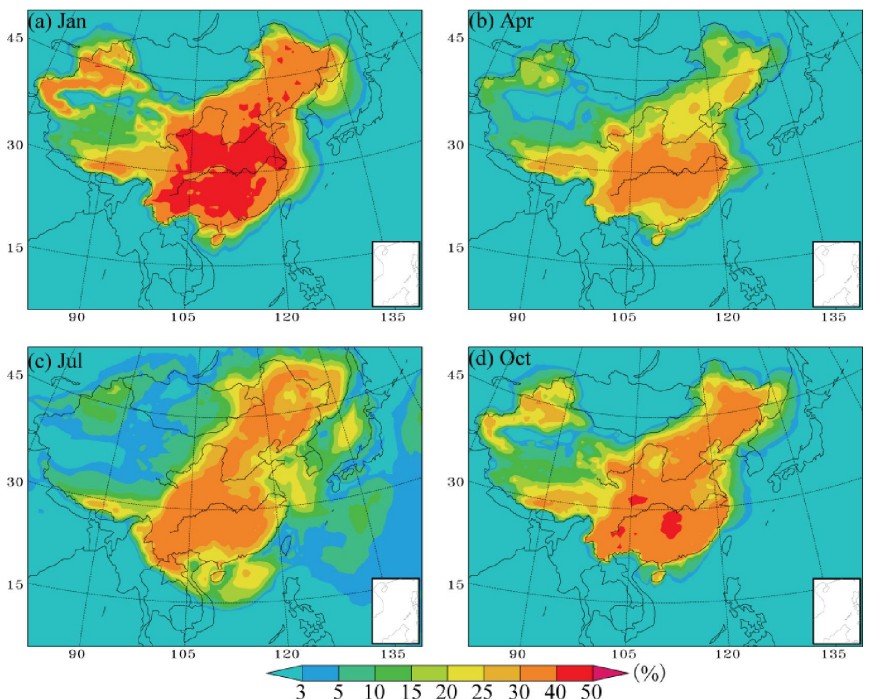


Figure 6. The horizontal distributions of the contribution percentage of NH$_3$ emissions to SNA mass concentration (%) in
January and July.


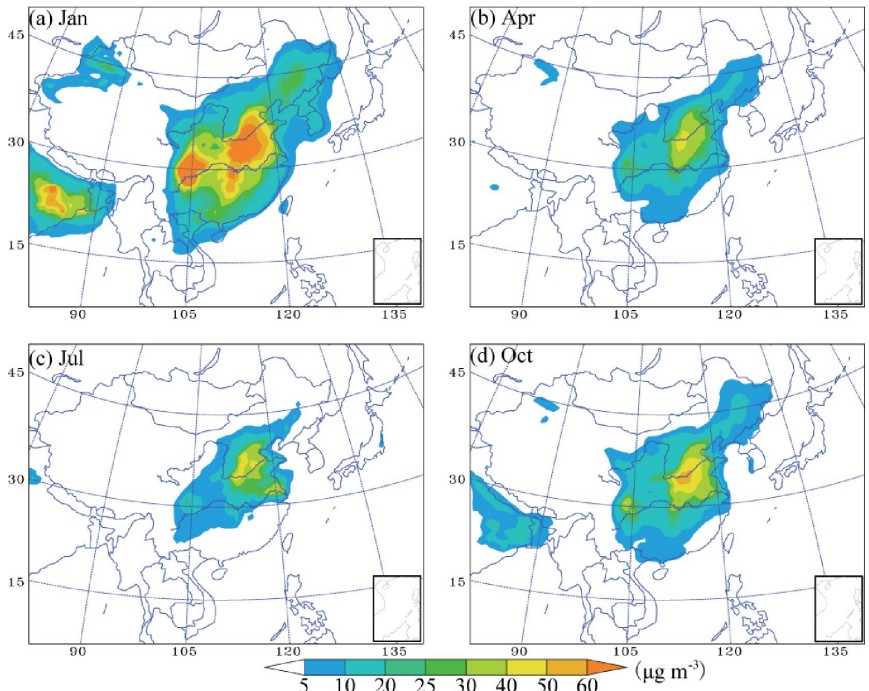


Figure 7. The horizontal distributions of SNA mass concentration ($\mu g\ m^{-3}$) variation associated with agriculture NH$_3$

removal in January and July.






















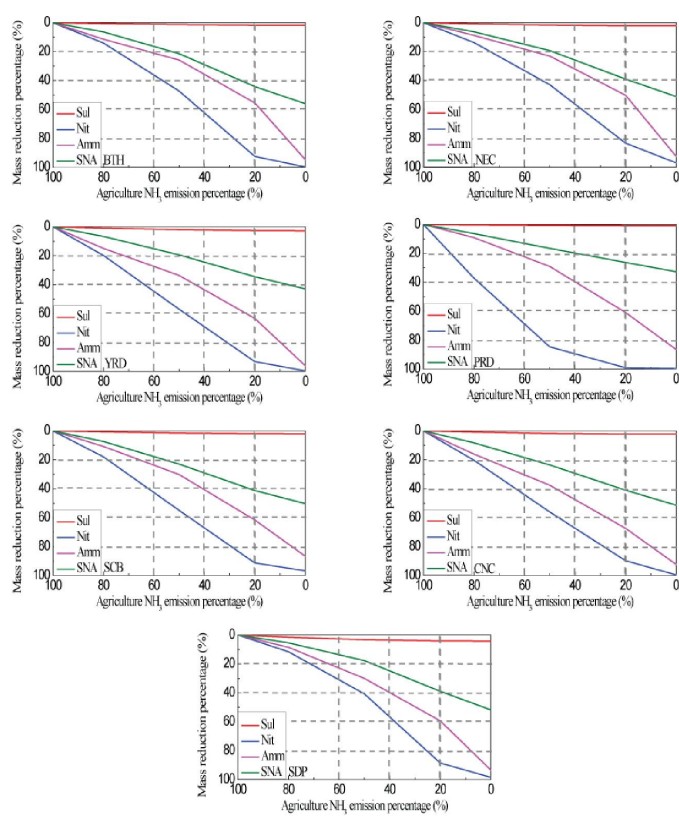

Figure 8. The variation (%) of sulfate, nitrate, ammonium, and SNA mass burden associated with the $NH_3$ emission
reduction (%).




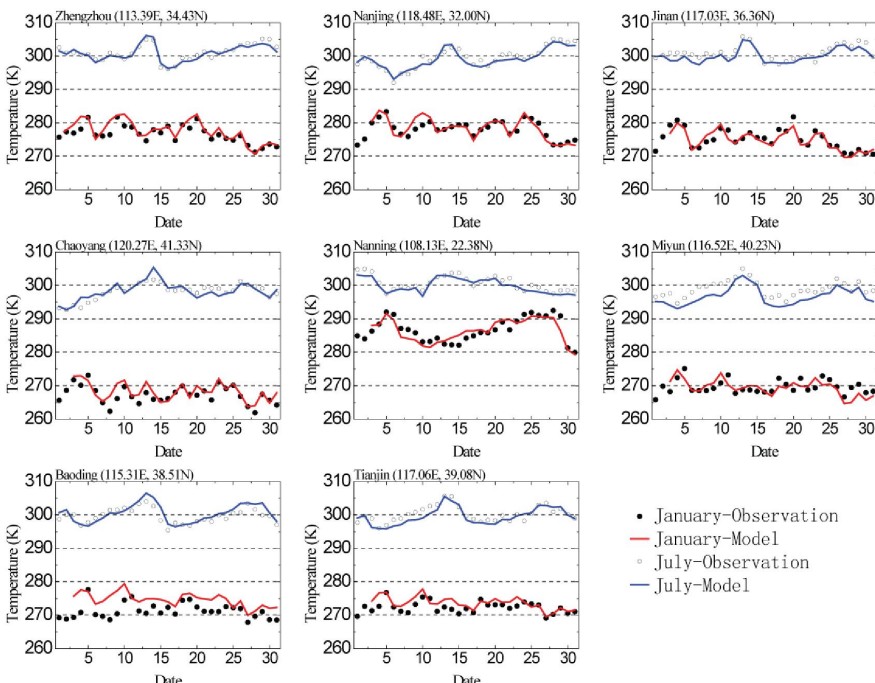

Figure A1. Observed and modeled daily average temperatures (K) in January and July 2015.



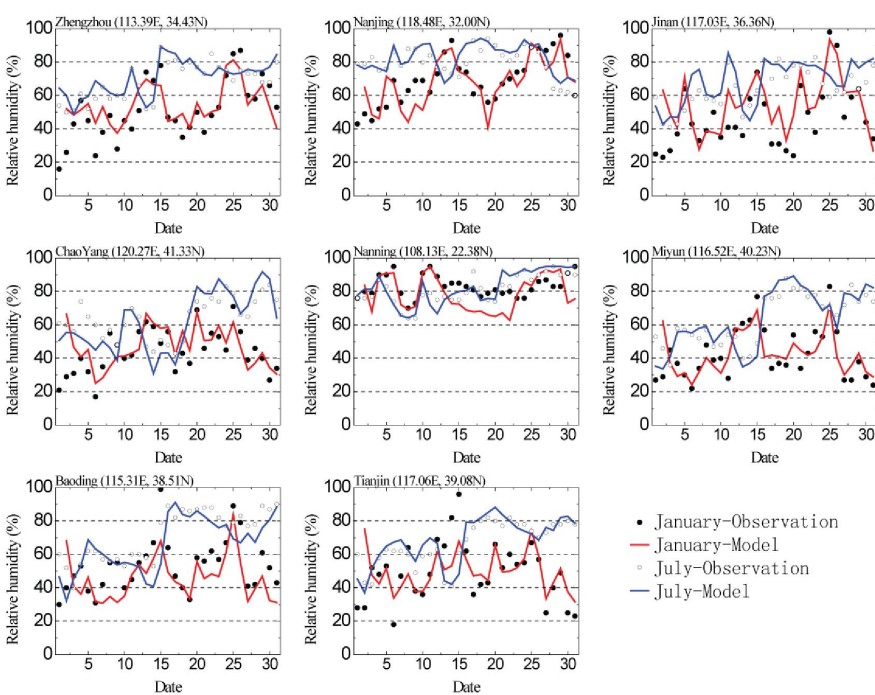

Figure A2. Same as Figure A1 but for relative humidity (%)





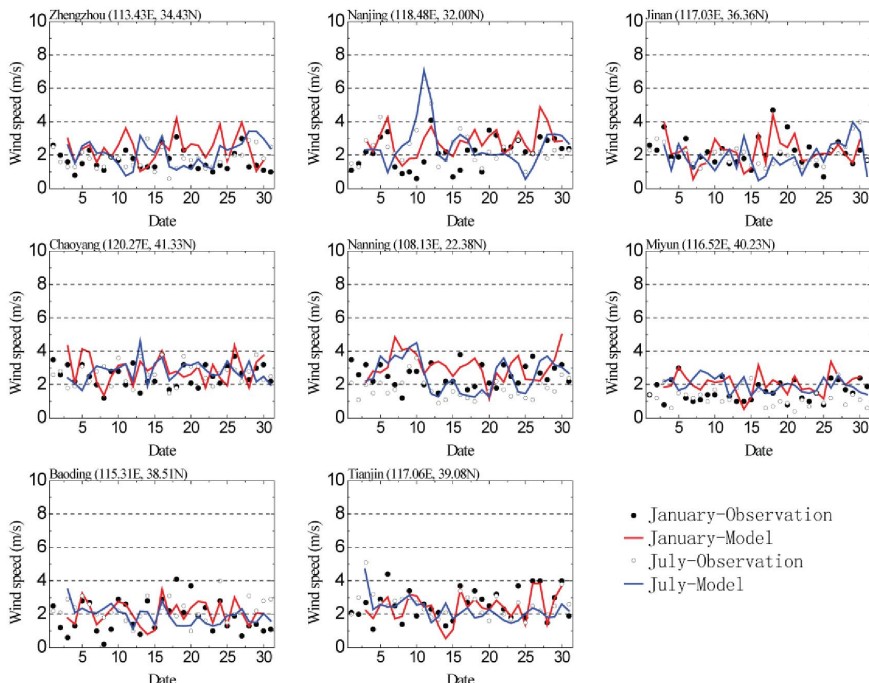

Figure A3. Same as Figure A1 but for wind speed (m s$^{-1}$)





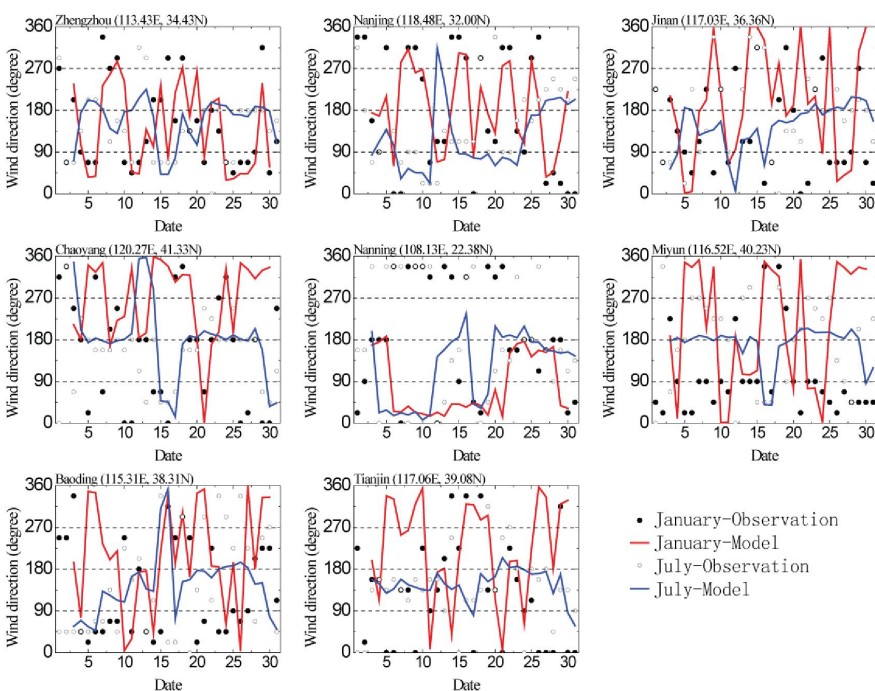

Figure A4. Same as Figure A1 but for daily maximum wind direction (degree)



Table 1. Statistical summary of the comparisons of the monthly average PM$_{2.5}$ between simulation and observation

|  |  | $N^a$ | $M^b$ | $O^c$ | $\sigma_m^d$ | $\sigma_o^e$ | $R^f$ | $FB^g$ | $NMB^h$ |
|---|---|---|---|---|---|---|---|---|---|
| PM$_{2.5}$ | Jan | 101 | 128.3 | 100.1 | 34.9 | 28.3 | 0.60 | 0.2 | 28.2 |
|  | Apr | 101 | 74.9 | 58.4 | 15.4 | 15.2 | 0.67 | 0.3 | 28.3 |
|  | Jul | 100 | 58.6 | 50.3 | 17.6 | 16.0 | 0.52 | 0.1 | 16.6 |
|  | Oct | 100 | 81.0 | 54.8 | 23.1 | 19.7 | 0.52 | 0.4 | 47.9 |
| NO$_2$ | Jan | 101 | 42.5 | 51.7 | 19.4 | 16.2 | 0.65 | -0.2 | -17.8 |
|  | Apr | 101 | 27.8 | 35.0 | 15.7 | 11.5 | 0.57 | -0.3 | -20.5 |
|  | Jul | 100 | 24.3 | 26.5 | 13.2 | 9.2 | 0.50 | -0.2 | -8.4 |
|  | Oct | 100 | 33.2 | 42.0 | 16.4 | 14.9 | 0.53 | -0.3 | -20.9 |
| SO$_2$ | Jan | 101 | 39.9 | 69.1 | 18.7 | 42.4 | 0.71 | -0.5 | -42.2 |
|  | Apr | 101 | 22.9 | 31.2 | 10.1 | 12.7 | 0.51 | -0.3 | -26.6 |
|  | Jul | 100 | 17.8 | 20.3 | 10.9 | 10.4 | 0.46 | -0.2 | -12.5 |
|  | Oct | 100 | 27.0 | 31.5 | 12.3 | 16.7 | 0.63 | -0.1 | -14.4 |

[a] Number of samples
[b] Total mean of observation
[c] Total mean of simulation
[d] Standard deviation of observation
[e] Standard deviation of simulation
[f] Correlation coefficient between daily observation and simulation
[g] Fractional Bias
[h] Nmalized Mean Bias



Table 2. The regional percent (%) of T contribution to sulfate, nitrate, ammonium, and SNA mass concentration.

|  | Sulfate | Nitrate | Ammonium | SNA | PM$_{2.5}$ |
|---|---|---|---|---|---|
| BTH | 1.1 | 8.0 | 83.3 | 31.9 | 15.5 |
| NEC | 1.0 | 5.6 | 83.7 | 28.1 | 14.3 |
| YRD | 1.0 | 7.4 | 85.7 | 29.2 | 15.3 |
| PRD | 0.9 | 5.8 | 90.6 | 33.5 | 14.2 |
| SCB | 0.7 | 5.1 | 93.9 | 32.6 | 13.0 |
| CNC | 0.9 | 6.0 | 92.8 | 36.6 | 17.5 |
| SDP | 0.9 | 7.1 | 80.5 | 30.1 | 15.1 |
| China | 2.2 | 10.1 | 87.6 | 29.0 | 16.0 |






Table 3. The variation percent (%) of sulfate, nitrate, ammonium, and SNA mass concentration associated with
agriculture NH₃ removal.

|  | Sulfate | Nitrate | Ammonium | SNA | PM₂.₅ |
|---|---|---|---|---|---|
| BTH | 0.7 | 99.8 | 94.7 | 49.4 | 34.4 |
| NEC | 0.7 | 96.9 | 92.5 | 48.9 | 31.1 |
| YRD | 5.0 | 99.2 | 96.1 | 48.8 | 31.6 |
| PRD | 2.0 | 99.2 | 97.2 | 40.3 | 23.4 |
| SCB | 2.6 | 96.7 | 85.9 | 49.8 | 25.9 |
| CNC | 1.9 | 99.2 | 92.3 | 50.9 | 32.3 |
| SDP | 2.7 | 99.5 | 93.4 | 46.6 | 34.0 |
| China | 1.6 | 98.8 | 93.8 | 45.7 | 25.2 |







