# Peer review of "Numerical analysis of the impact of agricultural emissions on PM2.5 in China using a"

_Atmospheric Chemistry and Physics, 2019_

## Referee Comment (RC1) · Anonymous Referee #2 · 10 Apr 2020

China is one of the largest agricultural countries in the world. The NH3 emissions from agricultural activities in China, such as fertilizer and husbandry, farmland ecosystems, livestock waste, crop residue burning and fuel wood combustion, significantly affect regional air quality and horizontal visibility by contribution to secondary inorganic aerosols. In the manuscript, the air quality modeling system RAMS-CMAQ (regional atmospheric modeling system-community multiscale air quality), coupled with the ISAM (integrated source apportionment method) module is applied to capture the contribution of NH3 emitted from total agriculture (Tagr) in China. It explores that the annual average contribution of Tagr NH3 to PM2.5 mass burden in China was 14-18%. Specific to the PM2.5 components, Tagr NH3 provided a major contribution to ammonium

formation (87.6%) but a tiny contribution to sulfate (2.2%). Though the Tagr NH3 only contributed 10.1% of nitrate under current emissions scenarios, the reduction of nitrate could reach 98.8% upon removal of the Tagr NH3 emissions. The results are meaningful, but the explanation for these phenomenon was not enough. I recommend the manuscript to be accepted after some minor revisions, and detail some issues below.

Major points: 1. The most important gas in this manuscript was NH3, but there are no NH3 in Figure 2 in comparing between the modeled and observed results. 2. Why is the NH3 contribution to nitrate small under "rich NH3" conditions and large in "poor NH3" environments? What is the internal logical relationship? 3. The study period is January, April, July, and October, but only the modeled and observed results in January and July are compared in Figure A1, A2, A3 and A4. 4. The author thinks that the obvious deviation between the observed and modeled SO2 in January may be a systemic underestimation due to the lack of emission intensity in this month. Did the lack of emission intensity only appear in SO2? Why are SO2 and NO2 underestimated and PM2.5 overestimated? 5. How much NH3 is removed in Figure 7? And it's more intuitive to use a negative value for reduction. 6. Why do the trend of the decrease in ammonium mass concentration accelerate while NH3 emissions is less than 20%? 7. What is the horizontal distributions of the contribution percentage of NH3 emissions to ammonium, nitrate and sulfate mass concentration, respectively? Which aerosol determines the horizontal distributions of SNA mass concentration? Why is the horizontal distributions of NH3 emissions different with the horizontal distributions of the contribution percentage of NH3 emissions to SNA mass concentration?

Minor points: 1. In Figure 6 and Figure 7, it should be the horizontal distributions in January, April, July, and October. 2. In Line 226, it should be "Since NH3 concerns mainly with secondary inorganic aerosols (SNA): sulfate, nitrate, and ammonium formation". 3. In line 269, what is "TA NH3 emission"? 4. In Line 833, should is it "The regional percent (%) of Tagr NH3 contribution"?

---

## Referee Comment (RC2) · Anonymous Referee #1 · 13 Apr 2020

The NH3 emissions from agricultural activities in China, which is one of the largest agricultural countries in the world, significantly affect regional air quality and horizontal visibility. In this study, the contributions of NH3 from multiple agricultural emissions to aerosols were calculated using the RAMS-CMAQ-ISAM system; it allowed to trace the transport and chemical reactions of NH3 from fertilizer and husbandry emissions sectors to quantitatively estimate the contribution of agricultural NH3 emissions to the PM2.5 mass concentration in China. As input was used the high-resolution PKU-NH3 emissions inventory, which was complemented with MIX Asian, REAS and GFED data; different meteorological factors were used to capture the formation processes and transport of secondary aerosols. For model evaluation, several observation data

were compared with the simulation results for both meteorological parameters and SO2, NO2, and PM2.5.

Major points. Suggestion: the "Results and discussions" section should be extended by providing explanations on different aspects. 1) How the emissions input influences the changes in concentrations patterns? Please discuss the seasonal variation in emissions for the months of January, April, July and October. 2) Identify, which of the agricultural sub-sectors, i.e., fertiliser, husbandry, farmland ecosystems, livestock waste, crop residue burning, and excrement waste from rural populations, are contributing most to the seasonal changes. 3) Emphasis the influence of meteorological conditions. 4) How this tool could support policy makers in designing the PM2.5 emissions mitigation strategy in China. 5) Explain why "the influence of NH3 would enhance with the decreasing of ambient NH3 mass concentration"; provide directions for further research on this topic.

Minor points. For the regions in China for which the findings are discussed – spell them out (e.g. "NEC"). Figure 6 – add to the caption "April and October". What is T in Table 2 - "of T contribution". What is TA, Page 9, line 269 - "10% TA NH3 emission".

---

## Author Comment (AC1) · 7 May 2020

China is one of the largest agricultural countries in the world. The NH3 emissions from agricultural activities in China, such as fertilizer and husbandry, farmland ecosystems, livestock waste, crop residue burning and fuel wood combustion, significantly affect regional air quality and horizontal visibility by contribution to secondary inorganic aerosols. In the manuscript, the air quality modeling system RAMS-CMAQ (regional atmospheric modeling system-community multiscale air quality), coupled with the ISAM (integrated source apportionment method) module is applied to capture the contribution of NH3 emitted from total agriculture (Tagr) in China. It explores that the annual

average contribution of Tagr NH3 to PM2.5 mass burden in China was 14-18%. Specific to the PM2.5 components, Tagr NH3 provided a major contribution to ammonium formation (87.6%) but a tiny contribution to sulfate (2.2%). Though the Tagr NH3 only contributed 10.1% of nitrate under current emissions scenarios, the reduction of nitrate could reach 98.8% upon removal of the Tagr NH3 emissions. The results are meaningful, but the explanation for these phenomenon was not enough. I recommend the manuscript to be accepted after some minor revisions, and detail some issues below. Major points:

1. The most important gas in this manuscript was NH3, but there are no NH3 in Figure 2 in comparing between the modeled and observed results.

R: Thanks for this comment. However, NH3 is not included in the conventional observation species in China at present. Therefore, it is hard to collect available observation data of NH3 mass concentration for model evaluation directly. Most of the available information was derived from the published research paper. In Han et al. (2017; Modeling dry deposition of reactive nitrogen in China with RAMS-CMAQ. Atmos. Environ.), the simulated NH3 by RAMS-CMAQ has been compared with the observations from many studies in detail, including the multi-year observation results with Nationwide Nitrogen Deposition Monitoring Network and the seasonal variation characteristics from Pan et al. (2012; Wet and dry deposition of atmospheric nitrogen at ten sites in Northern China. Atmos. Chem. Phys.). In this paper, we also compare the simulation results with the value and seasonal variation at several stations from Pan et al. (2018) and Zhang et al. (2018) (Line 200-211). We kindly hope these content could reflect the reasonability of modeled NH3.

2. Why is the NH3 contribution to nitrate small under "rich NH3" conditions and large in "poor NH3" environments? What is the internal logical relationship?

R: Thanks for this comment. In fact, the detail discuss about "rich NH3" and "poor NH3" can be found in Wang et al. (2011; Impact Assessment of Ammonia Emissions on

[Figure]

Inorganic Aerosols in East China Using Response Surface Modeling Technique). The results of RSM (Response Surface Modeling) in their study shows that the change of NO3- mass concentration is very sensitive to the emission level of NH4+ and performs as nonlinear relationship. The reduction of NH3 emissions can play a significant role in reducing the mass concentration of NO3- under NH3-poor condition. However, there will be excess NH3 in the atmosphere under NH3-rich condition, and these excess NH3 could neutralizes more nitric acid even in the case of emission reduction. Thus, the effect of emission reduction is not significant under NH3-rich condition. In addition, the SO2 will compete for NH3 and prevent the generation of NH4NO3 under NH3-poor condition because the reaction between H2SO4 and NH3 takes precedence over the one between HNO3 and NH3. Oppositely, SO2 should be benefit for the formation of NO3- (especially in summer) under NH3-rich condition according to the calculation of Wang et al. (2011). This should be a reason why the effect of NH3 emission control is not obvious in the case of NH3-rich condition as well.

3. The study period is January, April, July, and October, but only the modeled and observed results in January and July are compared in Figure A1, A2, A3 and A4.

R: Thanks for this comment. We added the comparison of meteorological factors in April and October. Please check if it is appropriate.

4. The author thinks that the obvious deviation between the observed and modeled SO2 in January may be a systemic underestimation due to the lack of emission intensity in this month. Did the lack of emission intensity only appear in SO2? Why are SO2 and NO2 underestimated and PM2.5 overestimated?

R: Thanks for this comment. The monthly mean observation data were used in the submitted version. However, we would like to provide more details about the evaluation. Thus, the hourly observation data from the China National Environmental Monitoring Centre were collected and compared with simulation results. The scatter plots (Figure 2) were replaced and the comparison of SO2, NO2 and PM2.5 in January, April,

July, and October at six sites were presented, and the statistical summary of the comparisons and related discussion were modified (Line 186-198). Please check if it is appropriate.

5. How much NH3 is removed in Figure 7? And it's more intuitive to use a negative value for reduction.

R: Thanks for this comment. Here the emission of NH3 from all agricultural sources were removed. For detail information, please see the percentage shown in Figure A6 which we added. In addition, the horizontal distribution of the PKU-NH3 emission inventory can be viewed in Kang et al. (2016) (Kang et al., 2016: High-resolution ammonia emissions, High-resolution, ammonia 1980, 2012.). On the other hand, the Figure 7 was modified. Please check if it is appropriate.

6. Why do the trend of the decrease in ammonium mass concentration accelerate while NH3 emissions is less than 20%?

R: Thanks for this comment. Here the simulation scenario was conducted for each emission reduction of 10% so that the acceleration should appear between 20% and 30%. In fact, it can be found that the accelerated decline mainly started when the emission reduction exceeds 50%. Therefore, we could deduce that the accelerated decline should be emerged gradually with NH3 emission reduction. This feature indicates that the formation of NH4+ should be nonlinear with NH3 emission intensity as well. The reason may also be related to the complex neutralization reaction among sulfate, nitrate and ammonium. The consumption of NH3 should become more sufficient when the mass concentration of NH3 is lower. Thus, the variation of ammonium is more sensitive under low NH3 mass burden.

7. What is the horizontal distributions of the contribution percentage of NH3 emissions to ammonium, nitrate and sulfate mass concentration, respectively? Which aerosol determines the horizontal distributions of SNA mass concentration? Why is the horizontal distributions of NH3 emissions different with the horizontal distributions of the

contribution percentage of NH3 emissions to SNA mass concentration?

R: Thanks for this comment. The horizontal distributions of NH3 emission contribution to sulfate, nitrate and ammonium is shown in Figure R1, and ammonium provided the major contribution to SNA (Table 4 also presented related information). In addition, Figure 6 shows the horizontal distributions of contribution percentage which may not follow the distribution pattern of mass concentration. For example, it can be seen that the agricultural NH3 emission generally provided more than 90% contribution to ammonium over China in January as shown in Figure R1. Therefore, the contribution ratio should differ from the horizontal distribution pattern.

Minor points: 1. In Figure 6 and Figure 7, it should be the horizontal distributions in January, April, July, and October. 2. In Line 226, it should be "Since NH3 concerns mainly with secondary inorganic aerosols (SNA): sulfate, nitrate, and ammonium formation". 3. In line 269, what is "TA NH3 emission"? 4. In Line 833, should is it "The regional percent (%) of Tagr NH3 contribution"?

R: Thanks for the comments. All error points were modified.

Please also note the supplement to this comment:
https://www.atmos-chem-phys-discuss.net/acp-2019-1128/acp-2019-1128-AC1-supplement.pdf
* * *

[Figure]

Figure R1 The horizontal distributions of the contribution percentage of NH₃ emissions to sulfate, nitrate and ammonium mass burden (%) in January and July.

**Fig. 1.** Figure R1 The horizontal distributions of the contribution percentage of NH3 emissions to sulfate, nitrate and ammonium mass burden (%) in January and July.

**Supplement:**

China is one of the largest agricultural countries in the world. The NH3 emissions from agricultural activities in China, such as fertilizer and husbandry, farmland ecosystems, livestock waste, crop residue burning and fuel wood combustion, significantly affect regional air quality and horizontal visibility by contribution to secondary inorganic aerosols. In the manuscript, the air quality modeling system RAMS-CMAQ (regional atmospheric modeling system-community multiscale air quality), coupled with the ISAM (integrated source apportionment method) module is applied to capture the contribution of NH3 emitted from total agriculture (Tagr) in China. It explores that the annual average contribution of Tagr NH3 to PM2.5 mass burden in China was 14-18%. Specific to the PM2.5 components, Tagr NH3 provided a major contribution to ammonium formation (87.6%) but a tiny contribution to sulfate (2.2%). Though the Tagr NH3 only contributed 10.1% of nitrate under current emissions scenarios, the reduction of nitrate could reach 98.8% upon removal of the Tagr NH3 emissions. The results are meaningful, but the explanation for these phenomenon was not enough. I recommend the manuscript to be accepted after some minor revisions, and detail some issues below.

Major points:

*1. The most important gas in this manuscript was NH3, but there are no NH3 in Figure 2 in comparing between the modeled and observed results.*

R: Thanks for this comment. However, $NH_3$ is not included in the conventional observation species in China at present. Therefore, it is hard to collect available observation data of $NH_3$ mass concentration for model evaluation directly. Most of the available information was derived from the published research paper. In Han et al. (2017; Modeling dry deposition of reactive nitrogen in China with RAMS-CMAQ. Atmos. Environ.), the simulated $NH_3$ by RAMS-CMAQ has been compared with the observations from many studies in detail, including the multi-year observation results with Nationwide Nitrogen Deposition Monitoring Network and the seasonal variation characteristics from Pan et al. (2012; Wet and dry deposition of atmospheric nitrogen at ten sites in Northern China. Atmos. Chem. Phys.). In this paper, we also compare the simulation results with the value and seasonal variation at several stations from Pan et

al. (2018) and Zhang et al. (2018) (Line 200-211). We kindly hope these content could reflect the reasonability of modeled $NH_3$.

*2. Why is the NH3 contribution to nitrate small under "rich NH3" conditions and large in "poor NH3" environments? What is the internal logical relationship?*

R: Thanks for this comment. In fact, the detail discuss about "rich $NH_3$" and "poor $NH_3$" can be found in Wang et al. (2011; Impact Assessment of Ammonia Emissions on Inorganic Aerosols in East China Using Response Surface Modeling Technique). The results of RSM (Response Surface Modeling) in their study shows that the change of $NO_3^-$ mass concentration is very sensitive to the emission level of $NH_4^+$ and performs as nonlinear relationship. The reduction of $NH_3$ emissions can play a significant role in reducing the mass concentration of $NO_3^-$ under $NH_3$-poor condition. However, there will be excess $NH_3$ in the atmosphere under $NH_3$-rich condition, and these excess $NH_3$ could neutralizes more nitric acid even in the case of emission reduction. Thus, the effect of emission reduction is not significant under $NH_3$-rich condition. In addition, the $SO_2$ will compete for $NH_3$ and prevent the generation of $NH_4NO_3$ under $NH_3$-poor condition because the reaction between $H_2SO_4$ and $NH_3$ takes precedence over the one between $HNO_3$ and $NH_3$. Oppositely, $SO_2$ should be benefit for the formation of $NO_3^-$ (especially in summer) under $NH_3$-rich condition according to the calculation of Wang et al. (2011). This should be a reason why the effect of $NH_3$ emission control is not obvious in the case of $NH_3$-rich condition as well.

*3. The study period is January, April, July, and October, but only the modeled and observed results in January and July are compared in Figure A1, A2, A3 and A4.*

R: Thanks for this comment. We added the comparison of meteorological factors in April and October. Please check if it is appropriate.

*4. The author thinks that the obvious deviation between the observed and modeled SO2 in January may be a systemic underestimation due to the lack of emission intensity in this month. Did the lack of emission intensity only appear in SO2? Why are SO2 and*

*NO2 underestimated and PM2.5 overestimated?*

R: Thanks for this comment. The monthly mean observation data were used in the submitted version. However, we would like to provide more details about the evaluation. Thus, the hourly observation data from the China National Environmental Monitoring Centre were collected and compared with simulation results. The scatter plots (Figure 2) were replaced and the comparison of $SO_2$, $NO_2$ and $PM_{2.5}$ in January, April, July, and October at six sites were presented, and the statistical summary of the comparisons and related discussion were modified (Line 186-198). Please check if it is appropriate.

*5. How much NH3 is removed in Figure 7? And it's more intuitive to use a negative value for reduction.*

R: Thanks for this comment. Here the emission of $NH_3$ from all agricultural sources were removed. For detail information, please see the percentage shown in Figure A6 which we added. In addition, the horizontal distribution of the PKU-$NH_3$ emission inventory can be viewed in Kang et al. (2016) (Kang et al., 2016: High-resolution ammonia emissions, High-resolution, ammonia 1980, 2012.). On the other hand, the Figure 7 was modified. Please check if it is appropriate.

*6. Why do the trend of the decrease in ammonium mass concentration accelerate while NH3 emissions is less than 20%?*

R: Thanks for this comment. Here the simulation scenario was conducted for each emission reduction of 10% so that the acceleration should appear between 20% and 30%. In fact, it can be found that the accelerated decline mainly started when the emission reduction exceeds 50%. Therefore, we could deduce that the accelerated decline should be emerged gradually with $NH_3$ emission reduction. This feature indicates that the formation of $NH_4^+$ should be nonlinear with $NH_3$ emission intensity as well. The reason may also be related to the complex neutralization reaction among sulfate, nitrate and ammonium. The consumption of $NH_3$ should become more sufficient when the mass concentration of $NH_3$ is lower. Thus, the variation of ammonium is more sensitive under low $NH_3$ mass burden.

*7. What is the horizontal distributions of the contribution percentage of NH3 emissions to ammonium, nitrate and sulfate mass concentration, respectively? Which aerosol determines the horizontal distributions of SNA mass concentration? Why is the horizontal distributions of NH3 emissions different with the horizontal distributions of the contribution percentage of NH3 emissions to SNA mass concentration?*

R: Thanks for this comment. The horizontal distributions of $NH_3$ emission contribution to sulfate, nitrate and ammonium is shown in Figure R1, and ammonium provided the major contribution to SNA (Table 4 also presented related information). In addition, Figure 6 shows the horizontal distributions of contribution percentage which may not follow the distribution pattern of mass concentration. For example, it can be seen that the agricultural $NH_3$ emission generally provided more than 90% contribution to ammonium over China in January as shown in Figure R1. Therefore, the contribution ratio should differ from the horizontal distribution pattern.

[Figure]

Figure R1 The horizontal distributions of the contribution percentage of $NH_3$ emissions to sulfate, nitrate and ammonium mass burden (%) in January and July.

Minor points:

d1. In Figure 6 and Figure 7, it should be the horizontal distributions in January, April, July, and October.

2. In Line 226, it should be "Since NH3 concerns mainly with secondary inorganic aerosols (SNA): sulfate, nitrate, and ammonium formation".

3. In line 269, what is "TA NH3 emission"?

4. In Line 833, should is it "The regional percent (%) of Tagr NH3 contribution"?

R: Thanks for the comments. All error points were modified.

---

## Author Comment (AC2) · 7 May 2020

The figures after modified in this paper are listed in the "supplement file".

Please also note the supplement to this comment:
https://www.atmos-chem-phys-discuss.net/acp-2019-1128/acp-2019-1128-AC2-supplement.pdf

---

## Author Comment (AC3) · 7 May 2020

The NH3 emissions from agricultural activities in China, which is one of the largest agricultural countries in the world, significantly affect regional air quality and horizontal visibility. In this study, the contributions of NH3 from multiple agricultural emissions to aerosols were calculated using the RAMS-CMAQ-ISAM system; it allowed to trace the transport and chemical reactions of NH3 from fertilizer and husbandry emissions sectors to quantitatively estimate the contribution of agricultural NH3 emissions to the PM2.5 mass concentration in China. As input was used the high-resolution PKUNH3 emissions inventory, which was complemented with MIX Asian, REAS and GFED data;

different meteorological factors were used to capture the formation processes and transport of secondary aerosols. For model evaluation, several observation data were compared with the simulation results for both meteorological parameters and SO2, NO2, and PM2.5. Major points. Suggestion: the "Results and discussions" section should be extended by providing explanations on different aspects.

1) How the emissions input influences the changes in concentrations patterns? Please discuss the seasonal variation in emissions for the months of January, April, July and October.

R: Thanks for this comment. The horizontal distribution, budget, and seasonal variations of PKU-NH3 emission inventory have been shown in the papers published by Pro. Song's research team (Kang et al., ACP, 2016; Liu et al, PNAS, 2019). Therefore, this information will not be displayed here again. However, we try to extract the data from PKU-NH3 emission inventory, and added Figure A5 which provided the regional average emission flux of each sectors (g/s/grid) over several major regions in January, April, July and October. It can be seen that the emission flux was higher in summer and lower in winter. The strongest emission flux mainly appeared in BTH, SDP and CNC, and the values in YRD, SCB and NEC was higher, too. These features generally followed the distribution pattern of NH3 mass concentration as shown in Figure 3. We have added the statement in Appendix A. Please check if it is appropriate.

2) Identify, which of the agricultural sub-sectors, i.e., fertiliser, husbandry, farmland ecosystems, livestock waste, crop residue burning, and excrement waste from rural populations, are contributing most to the seasonal changes.

R: Thanks for this comment. Furthermore, the contribution percent of each sector emission fluxes were calculated and shown in Figure A6. It can be seen that the highest proportion was contributed by husbandry, followed by the contribution of fertilizer. The total percent of husbandry and fertilizer was relatively higher in spring and summer, but lower in winter (brodly higher than 60% at least). In general, the emission

from husbandry and fertilizer should be the major contributor to NH3. In addition, the contribution of other sector, including industry, residential and transport, was also obvious. The discussion has been added to the paper (Appendix A). Please check if it is appropriate.

3) Emphasis the influence of meteorological conditions.

R: Thanks for this comment. We added the monthly horizontal distribution of the surface wind field in Figure 4, and modified related discussion (Line 225-227). Please check if it is suitable.

4) How this tool could support policy makers in designing the PM2.5 emissions mitigation strategy in China.

R: Thanks for this comment. The model system can be used to capture the source contribution features over the regions we concerned. The emission sectors and transport features can be obtained quantitatively based on the simulation results. Then, we can determine whether the control strategy is needed, and how much emission flux should be reduced. This is a useful tool for PM2.5 because most of the PM2.5 components are secondary species, which is hard to capture the source contribution features due to the strong nonlinear effect. Therefore, the model system should be helpful to make the PM2.5 emission control policy in China. However, the specific policies depends on more detail information in different regions (such as natural background, economic conditions, industrial structure, etc.), not only depends on the model simulation results.

5) Explain why "the influence of NH3 would enhance with the decreasing of ambient NH3 mass concentration"; provide directions for further research on this topic.

R: Thanks for this comment. This feature was deeply discussed by Wang et al. (2011; Impact Assessment of Ammonia Emissions on Inorganic Aerosols in East China Using Response Surface Modeling Technique). The results of RSM (Response Surface Modeling) in their study shows that the change of NO3- mass concentration is very

sensitive to the emission level of NH4+ and performs as nonlinear relationship. The reduction of NH3 emissions can play a very significant role in reducing the mass concentration of NO3- under NH3-poor condition. However, there will be excess NH3 in the atmosphere under NH3-rich condition, and these excess NH3 could neutralizes more nitric acid even in the case of emission reduction. Thus, the emission reduction effect is not significant under NH3-rich condition. In addition, the SO2 will compete for NH3 and prevent the generation of NH4NO3 under NH3-poor condition because the reaction between H2SO4 and NH3 takes precedence over the one between HNO3 and NH3. Oppositely, SO2 should be benefit for the formation of NO3- (especially in summer) under NH3-rich condition according to the calculation of Wang et al. (2011). This should be a reason why the effect of NH3 emission control is not obvious in the case of NH3-rich condition as well.

Minor points. For the regions in China for which the findings are discussed – spell them out (e.g. "NEC"). Figure 6 – add to the caption "April and October". What is T in Table 2 - "of T contribution". What is TA, Page 9, line 269 - "10% TA NH3 emission"

R: Thanks for the comments. All error points were modified.

Please also note the supplement to this comment:
https://www.atmos-chem-phys-discuss.net/acp-2019-1128/acp-2019-1128-AC3-supplement.pdf